# Identification of a mimotope of a complex gp41 human immunodeficiency virus epitope related to a non-structural protein of *Hepacivirus* previously implicated in Kawasaki disease

Hakimuddin Sojar,[1] Sarah Baron,[1] Mark D. Hicar[1]

**ABSTRACT**    Current HIV vaccine strategies are hampered by difficulty with recapitulating heavily mutated broadly neutralizing antibodies. We have previously isolated a highly mutated antibody termed "group C 76-Q13-6F5" (6F5) that uses immunoglobulin heavy chain variable region (VH)1-02. 6F5 targets a conformational epitope on HIV gp41 and mediates Ab-dependent cell cytotoxicity (ADCC). Reverting the group C 76 antibodies' variable chain to VH1-02 germline in antibody 76Canc showed retained ADCC activity. A vaccine targeting an epitope functionally recognized by germline antibodies offers a distinct advantage. Due to the 76Canc germline antibody ability to retain anti-HIV function, we sought to identify a protein target that could form the basis of a vaccine. 76Canc specifically recognized a number of acidic peptides on a microarray containing 29,127 linear peptides. Meme analysis identified a peptide sequence similar to a non-structural protein of *Hepacivirus* previously implicated in Kawasaki disease (KD). Binding was confirmed to significant peptides, including the *Hepacivirus*-related and KD-related peptide. On serum competition studies using samples from children with KD compared to controls, targeting of this epitope showed no specific correlation to the clinical syndrome of KD. Yeast-displayed human protein microarray autoantigen screening was also reassuring. This study identifies a peptide that can mimic the gp41 epitope targeted by 76C group antibodies (i.e., a mimotope). We show little risk of autoimmune targeting inclusive of inflammation similar to KD, implying non-specific humoral immunity targeting of similar peptides during KD. Development of an HIV vaccine based on such peptides should proceed, but with continued caution.

**IMPORTANCE**   The development of protective HIV vaccines continues to remain a significant challenge. Many of the broadly neutralizing antibodies require a significant number of mutations, suggesting that traditional vaccines will not be able to recapitulate these types of responses. We have discovered an antibody that has Ab dependent cell cytotoxicity (ADCC) activity against HIV even when mutating the heavy chain of that antibody to germline. As a potential target for vaccines, this offers a distinct advantage: a few immunizations should directly stimulate B cells harboring those specific germline variable chains for expansion. This study sought to identify potential peptide targets that could be formulated into such a vaccine. We identified a peptide that both germline and mature antibodies can recognize. Initial autoantigen screens and consideration of inflammatory disorders suggest this identified antigen is a feasible approach to move forward into pre-clinical models.

**KEYWORDS**    HIV, ADCC, Kawasaki disease, peptide mimotope, gp41 antibody, Hepatitis C

**Peer Reviewer** Arden Baylink, Washington State University, Pullman, Washington, USA

Address correspondence to Mark D. Hicar, markhica@buffalo.edu.

The authors declare no conflict of interest.

See the funding table on p. 11.

The creation of a successful human immunodeficiency virus (HIV) vaccine continues to be a public health priority (1). A large effort has been focused on discovery and characterization of broadly neutralizing antibodies (bnAbs) (2, 3). Many bnAbs are highly mutated, but increased levels of mutations can be stochastic and do not predict neutralization. The increased number of mutations in broadly neutralizing antibodies suggests that traditional vaccines will not be able to recapitulate these types of responses.

The monoclonal antibody (Ab) 76-Q13-6F5 (6F5) is highly mutated (83% homologous to predicted heavy chain germline), and has the capacity to mediate Ab-dependent cell cytotoxicity (ADCC) (4). The 6F5 epitope encompasses areas in both heptad repeats of gp41, mapped by alanine scanning mutagenesis to amino acids (AA) R557, E654, and E657 of reference sequence HXB2, just proximal to the membrane-proximal external region (MPER-underlined) (5). Three other Abs (76-Q11-4E4, 76-Q7-6F11, and 76-Q7-7C6) used variable region (VH)1-02, competed for the 6F5 binding, and were also shown to target E657 AA (bold) (AA 652-667: QQEKN**E**QELL<u>ELDKWA</u>) (5, 6). In prior studies, we termed these four antibodies 76C Abs or group C Abs.

Serum from HIV long-term non-progressors contained significantly higher levels of group C Abs in comparison to HIV-infected persons with comparable viral loads. Due to this correlation with non-progression, further studies were done by creating a 76C group ancestor (76Canc) Ab utilizing the unmutated germline heavy variable chain from VH1-02. From exploring the derivation and possible cross-reactivity of 76Canc, we discovered this ancestor Ab also has significant functional ADCC activity (4).

Development of Abs utilizing VH1-02 gene segments after vaccination is well studied, as VH1-02 is used in VRC01, one of the most broadly neutralizing of Abs (7, 8). VRC01 is highly mutated, with V-gene region AA predicted mutations of 42% in the heavy chain and 28% in the light chain. The recognition of the CD4 binding site by VRC01 is predominantly driven by this VH1-02 heavy chain (7) and related Abs rely on similar structures (8, 9). Neutralization was maintained when VRC01 framework mutations were mutated closer to germline (10), but completely unmutated common ancestors of VRC01 and related Abs don't interact with native trimers. This lack of binding to germline creates a challenge for vaccination strategies attempting to stimulate the naïve Ab repertoire to generate an HIV bNab response (8, 11–14). An epitope that could be targeted functionally by Abs using germline variable chains would offer an advantage as a vaccine candidate.

As we have shown that Abs related to 6F5 correlate with non-progression and that germline use of VH1-02 in 76Canc can support anti-HIV functional ADCC, we propose a vaccine strategy to create such 76Canc-like Abs. Unfortunately, a number of studies utilizing gp41 constructs, including trimeric forms, have been relatively unsuccessful (15). Since numerous vaccine strategies with recombinant gp41 have not been successful, we sought to discover a specific peptide target that could be recognized by 76C group Abs, including the ancestor Ab 76Canc, to be used in future immunization studies. Ideally, this target would not be implicated in other inflammatory conditions and would have little cross-reaction to human autoantigens.

## MATERIALS AND METHODS

### Enrollment

Plasma samples from febrile children, including Kawasaki disease (KD) subject samples (UBKD) and associated clinical information were collected under approval of the UB IRB STUDIES- 00000126, 00002824, and 00005262 with funding support by the Wilder-muth Memorial Foundation as previously described (16). Additional serum samples (30 complete KD subjects with pre-intravenous immunoglobulin (IVIG) treatment, post-IVIG, and convalescent samples) were obtained through the Pediatric Heart Network and stored in the Kawasaki Disease Biorepository at Boston Children's Hospital (IRB X10-01-0308), which were collected for a prior study (17). Statistical analysis was

performed using GraphPad Prism 9, and groups were compared with Wilcoxon ranked sum tests.

## Serum antigen targeting screening

Serum samples were provided to CDI laboratories to screen on the HuProt array. The HuProt array is a yeast-derived expression library of 23,059 human proteins. These targets are duplicated on the screens, and binding is normalized to background binding and calculated per company's protocols. Specific Abs were screened per company protocols on the PEPperCHIP Human Epitome Microarray, containing 29,127 linear peptides printed in duplicate. The peptide content was based on all linear B-cell epitopes of the Immune Epitope Database with the host "human" and was further complemented by all epitopes of the most common vaccines.

## Meme analysis

The top 65 Ab targets identified on the PEPperCHIP Human Epitome Microarray (>200 fluorescence units threshold) were uploaded to the MEME tool (http://meme-suite.org//tools/meme). The MEME pre-settings were a maximum of one motif per each sequence with a maximum total of five different motifs, as well as a minimum motif length of 4 AA and a threshold of E < 5.0e-002.

## Peptide ELISA and characterization

Peptide enzyme-linked immunosorbent assays (ELISAs) proceeded as previously described (18) with the following adjustments: peptides were dissolved in 50% DMSO in PBS and coated at 10 ng/well of peptide and incubated overnight at 4℃ on a rocking platform prior to assay. For biotinylated Ab competition ELISAs, Ab biotinylation, and ELISA were performed as previously described (4). Absorbance was read on the BioTek Synergy LX Multi-Mode Microplate Reader and is reported as optical density (OD) per manufacturer. Peptide characteristics (isoelectric point, charge at pH 7, and hydrophilicity) were calculated with online calculator (Bachem.com) with N-terminal -H and C-terminal -OH. Assays were run thrice with representative results shown unless otherwise noted.

## Protein binding ELISA: confirmation of autoantigen targeting

For western blotting and ELISA assays, human glutaredoxin 3 (GLRX3, catalog # TP302731), and human Tropomodulin 1 (TMOD1, catalog # TP301134) were obtained from OriGene Technologies Inc., Rockville, MD. TMOD1 human recombinant isoform 1 (NP_003266.1) and GLRX3 isoform 1 (NP_006532.2) were used in BLAST analysis. Recombinant protein ELISAs proceeded as previously described (19) with the following adjustments: proteins were plated at 10 ng/well overnight at 4℃, for GLRX3, 1% BSA was used as diluent, and for TMOD1, 7.5% FBS in PBS was used as diluent.

## Western blot analysis

For the slot blot, peptides were transferred onto blotting membrane using the Bio Dot Microfiltration system (Bio Rad Chemical Cat#170398) according to the manufacturer's instructions. Western blots were blocked with 1% BSA in pH 7.5 Tris-Buffer saline for 1 h at room temperature. After rinsing, the primary Ab was diluted in 1% BSA in Tris-Buffer saline pH 7.5 and incubated overnight at 4℃. The blot was washed (3 × 10 min) with gentle agitation. Secondary Ab (Alkaline phosphatase-conjugated anti-human IgG, Southern Biotech, Birmingham, Al) was added in 1% BSA in Tris-Buffer saline pH 7.5 and incubated for 1 h at room temperature with gentle agitation. Blot was then washed three times in Tris buffer saline pH 7. Bands were visualized with alkaline phosphate substrate NBT/BCIP (Thermo Scientific, Grand Island, NY). Assays were run thrice with representative results shown unless otherwise noted.

## RESULTS

From prior studies, the epitope targeted by the group 76C Abs is conformational (6), but numerous vaccine strategies using recombinant gp41 have not been successful (15). Further definition of the 76C Ab epitope can be facilitated by isolating peptides that can replicate such discontinuous conformational epitopes; or so-called mimotopes (10). We interrogated the PEPperCHIP Human Epitome Microarray, covering 29,127 linear peptides, to search for possible mimotopes. The peptide content was based on all linear B-cell epitopes of the Immune Epitope Database with the host "human," and was further complemented by all epitopes of the most common vaccines.

On library screening using 76Canc, which has the unmutated VH1-02 segment, the most significant binding was against a number of negatively charged peptides from glycinin (*Arachis hypogaea*) with the consensus motif EYDEDEYEY (Fig. 1). Most of the top hits were highly acidic with the average calculated isoelectric point of the top 20 being 3.05 with charge at neutral pH of −6.84. This is not surprising since the 76 group C epitope is in an acidic hydrophilic region in the carboxy-terminal heptad repeat (Hxb2 gp160 reference AA 652-667: QQ**E**KN**E**QELLELDKWA; bolded/underlined resolved by alanine scanning mutagenesis [6]). Numerous possible human pathogen motifs were identified, with many of these being negatively charged. A number of human peptides enriched for negatively charged acidic AA were also readily recognized from the cerebellar degeneration-related antigen 1, Major Centromere Autoantigen B, and coagulation factor VIII precursor.

No HIV-related peptides showed significant binding activity (detailed in Table S1), consistent with the lack of gp41-derived peptide binding in prior studies (6). This included six HIV peptides that overlapped with the 76C group E657 motif (red text, Table S1), all of which had minimal binding (<50 relative binding units), including the very acidic peptide EELKQLLEQWNLVIGFL (i.e., 3.95). As HIV and coronaviruses (CoVs) both utilize type 1 fusion proteins to infect cells, it is plausible there could be cross-reactive Abs that may target a structural domain on the fusion proteins (HIV envelope and CoV Spike). Peptides derived from SARS-CoV were included in the peptide screen and showed no appreciable binding, including the Spike S2 peptide PLKPTKRSFIEDLLF, which is homologous to the 76C group epitope on gp41.

## Meme analysis

From the identified peptides, we sought to identify an optimal target to move forward with in a vaccine. To explore consensus targets, a MEME analysis of all peptides with a spot intensity of >200 fluorescence units (top 65 hits, details in Table S1) was performed (Fig. 2). The top motif exhibited a very high statistical significance of E = 5.3e-083 with contributions from 10 of 65 top hits and a motif length of 17 AA. This motif mainly

| Peptide | Relative binding units | Organism | Protein | IsoElectric Point | Charge pH 7 | Hydrophilicity |
|---|---|---|---|---|---|---|
| DEEEEYDEDEYEYDE | 2,444.0 | *Arachis Hypogaea* | Glycinin | 2.45 | -11.99 | 1.94 |
| EEEEYDEDEYEYDEE | 2,018.5 | *Arachis Hypogaea* | Glycinin | 2.47 | -11.99 | 1.94 |
| RADEEEEYDEDEYEY | 1,986.0 | *Arachis Hypogaea* | Glycinin | 3.12 | -8.99 | 1.71 |
| EEYDEDEYEYDEEDR | 1,751.0 | *Arachis Hypogaea* | Glycinin | 3.08 | -9.99 | 1.94 |
| YVRQLEQYFDNFDQDFL | 1,173.0 | *Plasmodium Vivax Sal-1* | Vacuolar Atp Synthase Catalytic Subunit A | 3.41 | -3.00 | -0.08 |
| FLEDVPWLEDVDFLED | 974.5 | *Homo Sapiens* | Cerebellar Degeneration-Related Antigen 1 | 2.57 | -6.99 | 0.26 |
| CDKNTGDYYEDSYED | 865.0 | *Homo Sapiens* | Coagulation Factor Viii Precursor | 3.23 | -5.04 | 0.88 |
| NEEAEDYDDDLTDSEMD | 789.0 | *Homo Sapiens* | Coagulation Factor Viii Precursor | 2.43 | -9.99 | 1.42 |
| VDHFADGYDE | 774.0 | *Aspergillus Fumigatus* | Major Allergen Asp F 2 Precursor | 3.41 | -3.91 | 0.47 |
| PVNDLCYPGDFNDYEEL | 771.5 | *Influenza A Virus H5N1* | Hemagglutinin | 2.69 | -5.04 | 0.13 |
| SFSKYVRQLEQYFDNFD | 689.0 | *Plasmodium Vivax Sal-1* | Vacuolar Atp Synthase Catalytic Subunit A | 4.41 | -1.00 | 0.05 |
| EYDEDEYEYDEEDRR | 602.0 | *Arachis Hypogaea* | Glycinin | 3.36 | -7.99 | 1.94 |
| DAWREGEEFVVEFDL | 577.5 | *Mycobacterium Leprae* | 18 Kda Antigen | 3.31 | -4.99 | 0.49 |
| EDYDDDLTDSEMDVVRF | 566.0 | *Homo Sapiens* | Coagulation Factor Viii Precursor | 3.1 | -6.99 | 0.94 |
| LSFSCYLSVTEQSEFYF | 537.0 | *Human Hepatitis A* | Genome Polyprotein | 3.09 | -2.04 | -0.66 |
| KNNEEAEDYDDDLTD | 471.5 | *Homo Sapiens* | Coagulation Factor Viii Precursor | 3.12 | -6.99 | 1.49 |
| DSEEEDDEEEDDEDE | 459.5 | *Homo Sapiens* | Major Centromere Autoantigen B | 2.36 | -13.98 | 2.82 |
| VIPDREVLYQEFDEMEE | 356.0 | *Hepatitis C Virus Subtype 1A* | Polyprotein | 3.26 | -5.99 | 0.68 |
| WVDHFADGYD | 338.0 | *Aspergillus Fumigatus* | Major Allergen Asp F 2 Precursor | 3.53 | -2.91 | 0.17 |
| LQSDQEEIDYDDTISVE | 335.0 | *Homo Sapiens* | Coagulation Factor Viii Precursor | 2.57 | -6.99 | 0.73 |
| Average values | | | | 3.05 | -6.84 | 0.96 |

**FIG 1** Top 20 peptides recognized by 76Canc. Relative binding units are shaded in comparison to zero, which is the normalized background. Peptide characteristics (isoelectric point, charge at pH 7, and hydrophilicity) were calculated using the online calculator (Bachem.com).

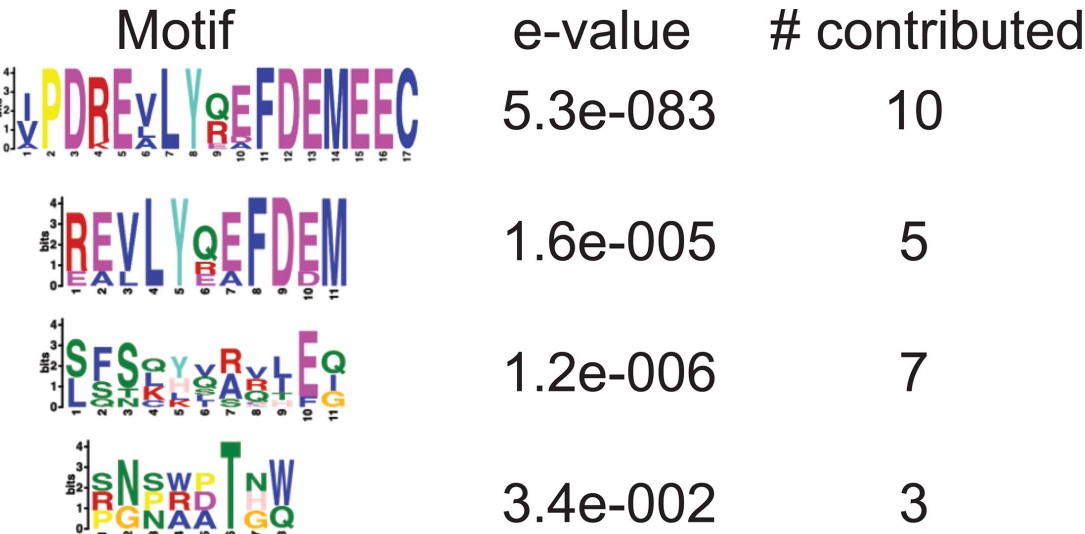

| Motif | e-value | # contributed |
|---|---|---|
| | 5.3e-083 | 10 |
| | 1.6e-005 | 5 |
| | 1.2e-006 | 7 |
| | 3.4e-002 | 3 |

**FIG 2** Motif meme analysis of top 65 peptides recognized by 76Canc. The top 65 identified peptides were analyzed using meme analysis, http://meme-suite.org//.

originated from various similar *hepatitis C virus* (HCV) peptides. Due to the uncommon epitope length, it's possible these peptides could replicate a conformational epitope. It's also possible the main motif was based on a shorter acidic portion of the C-terminal, as the second motif (Fig. 1, REVLYxxFDEM) was a shorter sequence within the first motif. It appears unlikely the FDEM sequence alone is targeted as there were 64 HCV FDEM-containing peptides in the screen, but only 15 with spot intensity of >200 fluorescence units (see Table S1).

## Confirmatory binding

Five peptides that reflected top peptide hits (Table 1) and the meme analysis were produced and compared to peptides from a number of pathogens of interest and an acidic peptide from *Plasmodium falciparum* that did not show appreciable binding on the peptide microarray. On the ELISA assay, biotinylated Abs of 76Canc, 6F5, and 6F11 bound all five top-hit peptides over the background of the control Ab (Fig. 3A). Other notable targets showed specific binding from the top 65 hits (outer membrane protein of *Neisseria meningitidis* and AA permease of *Staphylococcus aureus*). A collection of acidic peptides (Table 1: 8, 9, and 10) and the blank well (50% DMSO only) were negative (peptide data 9 and 10 not shown). A slot blot assay was performed and confirmed binding to a number of these peptides, roughly corresponding to the level over background in ELISA results (Fig. 3B; Table 1).

The microarray contains over 5,000 peptides from human proteins. A number of human peptides were in the top 65: Cerebellar Degeneration-Related Antigen 1, Coagulation Factor Viii Precursor, Major Centromere Autoantigen B, Kinesin-Like Protein Kif11, 78 kDa Glucose-Regulated Protein, Glutamate Decarboxylase 2, Calcium Channel, Alpha 1A Subunit Isoform 3, Heat Shock Protein 90Kd, DNA-Directed RNA Polymerase Iii Subunit, Rpc1, Trinucleotide Repeat Containing 6A, and Isoform CraB Envoplakin. We expressed the top human peptide (Table 1, peptide #3), which showed binding over twice background on ELISA (Fig. 3), but was not shown to bind on slot blot analysis.

## Hepatitis C virus-related peptide

The HCV-related peptide identified herein is similar to a recently identified peptide implicated in relation to KD (20), KPAVIPDREALYQDIDEMEEC. This peptide was identified in prior published studies using the Ab KD4-2H4, a plasmablast-derived antibody from a child with KD. This KD4-2H4 peptide was related to a sequence from a non-structural

**TABLE 1** Selected peptides from the peptide microarray screen produced for confirmation[a]

| # | Peptide | IP | pH7 charge | Protein target | Organism | WB | Notations |
|---|---------|-----|-----------|----------------|----------|-----|-----------|
| 1 | ADEEEEYDEDEYEYDEEDR | 3.02 | −12.98 | Ara H3 allergen, glycinin | *Arachis hypogaea* | Yes | Top peptide, Fig. 1 |
| 2 | VIPDREVLYQEFDEMEE | 3.26 | −5.99 | Non-structural protein | *Hepatitis C Virus* | Yes | Top two meme motifs related |
| 3 | NEEAEDYDDDLTDSEMD | 2.43 | −9.99 | Coagulation Factor VIII Precursor | *Homo sapien* | No | Top human peptide related, Fig. 1 |
| 4 | VDHFADGYDE | 3.41 | −3.91 | Major Allergen Asp F 2 Precursor | *Aspergillus fumigatus* | No | Top 10 peptide, acidic, Fig. 1 |
| 5 | YVRQLEQYFDNFDQDFL | 3.41 | −3.00 | Vacuolar Atp Synthase Catalytic Subunit A, Putative | *Plasmodium vivax Sal-1* | Yes | Top 10 peptides related to the third meme motif |
| 6 | EYDQVVGAE | 2.93 | −3.00 | Serotype 15 Outer Membrane Protein | *Neisseria meningitidis* | Yes | Acidic |
| 7 | SFNLLSARLFGELFW | 6.99 | 0 | Amino Acid Permease | *Staphylococcus aureus* | Yes | Neutral |
| 8 | APSVEESVAPSVEESVA | 2.95 | −3.99 | Liver Stage Antigen-3 | *Plasmodium falciparum* | No | Acidic, low level of binding on screen; control on slot blot |
| 9 | AYDKDRYTEEEREVYSY | 4.16 | −3.00 | Skc-2 | *Streptococcus dysgalactiae* | np | 197 immunofluorescent units on microarray |
| 10 | SQGISDDDNDSAVAEFF | 2.64 | −5.00 | Genome Polyprotein | Human hepatitis A | np | 197 immunofluorescent units on microarray |

[a]IP, isoelectric point; pH7charge, net charge at pH 7.0; WB, binding on slot blot; np, not performed.

protein of HCV. KD is a vasculitis of children thought to be related to an infectious disease (21, 22). Despite an extensive history of studies attempting to associate an infection with KD, the cause of KD remains unknown (23). The specificity of binding was assay dependent, as there appeared to be binding by immunohistochemistry, but high concentration of Ab was needed to show appreciable KD4-2H4 peptide binding in ELISA (>1 ug/mL) (20).

We compared the binding of KD4-2H4 peptide to Peptides #1-3 from Table 1. We show that the binding of 6F5 and 6F11 readily recognizes all of these peptides with diminished binding by the 76Canc ancestor compared to the HIV 6F5 and 6F11 Abs (Fig. 4). Notably, reviewing the history of subject 10076 from whom these Abs were originally derived (5, 6), it should be noted this subject did not report a peanut allergy and was repeatedly negative on HCV testing.

## Clinical correlations

It remains unclear how aneurysms form during KD, although targeting by autoantibodies (autoAbs) is a proposed mechanism (22, 24, 25). Since these 76C group Abs readily recognize KD4-2H4 peptide, we sought to assess if there was a correlation in 76C group Abs to KD. The 6F11 Ab was biotinylated, and serum from a cohort of children with KD and febrile controls were used. Competitions of serum to 6F11 binding to KD4-2H4 peptide showed no differences (Fig. 5, Mann-Whitney $P = 0.44$) between KD and febrile controls. We additionally assessed a cohort of 30 children with complete KD, with serial pre-IVIG, post-IVIG, and convalescent samples, as previously described (26). Overall, there was not a significant increase in KD4-2H4 peptide-targeting Abs that occurred in convalescent KD samples (pre-IVIG vs convalescent sample Mann-Whitney $P$ value 0.77). After IVIG administration, there was no appreciable dilutionary effect in the majority of individuals, implying most IVIG formulations already contain Abs that would similarly bind this antigen. In subgroup analysis comparing those with elevated coronary artery Z scores, there was no overall difference between those with or without aneurysms for the pre-IVIG, post-IVIG, and convalescent comparisons (Mann-Whitney $P = 0.71$, $P = 0.41$, and $P = 0.24$).

## Autoimmune assessment

To further assess if this would be a safe vaccination strategy, we utilized the HuProtTM library (CDI Labs), a yeast-derived expression library consisting of 23,059 purified human proteins to further assess potential autoimmunity. We compared the binding of 6F5

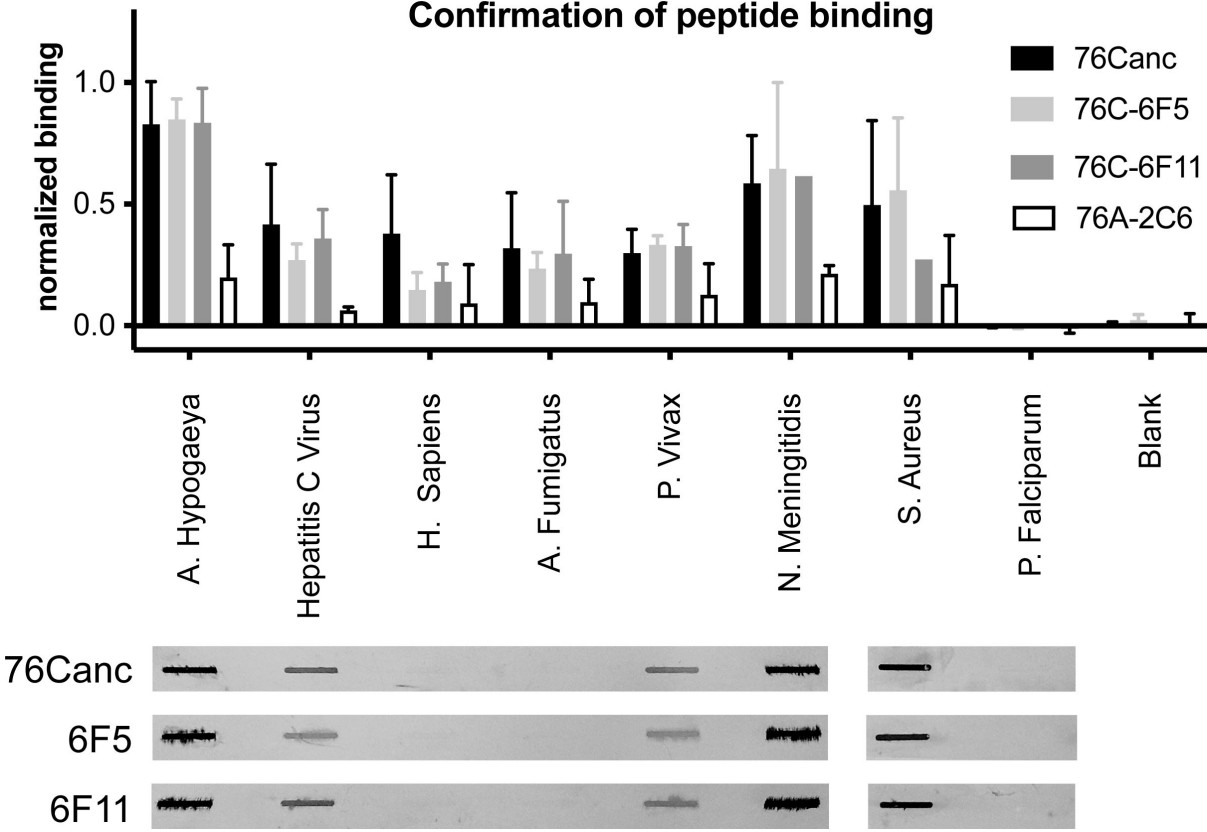

**FIG 3** Binding assays of group C Abs to peptides representing meme analysis. (A) ELISA was used to assess binding to peptides representing top hits and various controls from the peptide screen. Mean and standard error are shown of three experiments using comparable parameters to the original peptide screen (5 ug/mL of Ab). (B) Peptide binding was assessed by western slot blot (results shown all from a single blot, image was arranged to align to the ELISA data).

Ab with the 76Canc (Fig. 6). Significance in this assay is defined both by the Z score difference from total background library protein binding and comparing to the next closest Z score result. 6F5 (Fig. 6A, gray dots) showed two significant findings, Glutaredoxin 3 (GLRX3, duplicated on the library) and microtubule associated scaffold protein 2 (MTUS2). Overall, 76Canc (Fig. 6A, black boxes) had generally less autoantigen binding over background than its more mature relative. Tropomodulin 1 (TMOD1, duplicated on the library) and, to a lesser extent, TMOD4 reached significance. Notably, none of these proteins were seen in the peptide array findings (Fig. 1).

To confirm the validity of this screening array, we chose the top hits to confirm binding. GLRX3 is a fairly acidic protein, with a theoretical PI of 5.31 and containing 14.6% acidic AA. On the array, median binding over background ratio was 83 and 26. TMOD1 also had a theoretical PI of 5.01 containing 16.7% acidic AA and had a median binding over background ratio on the array of 443 and 462.

ELISA testing on recombinant GLRX3 and TMOD1 showed similar modest binding patterns as shown in the array. Western blot analysis showed that 76Canc readily recognized TMOD1 on denatured gels (Fig. 6C) and native dot blot (Fig. 6D). Binding of 6F5 to GLRX3 was more difficult to resolve on both modalities and showed variable results consistent with a lower affinity interaction. BLAST alignment did not reveal significant homology between TMOD1 and GLRX3. A number of acidic regions in TMOD1 were present that aligned to peptides identified in our study (Supplemental 2), further suggesting the acidic nature of these epitopes may be contributing to this cross-reactivity.

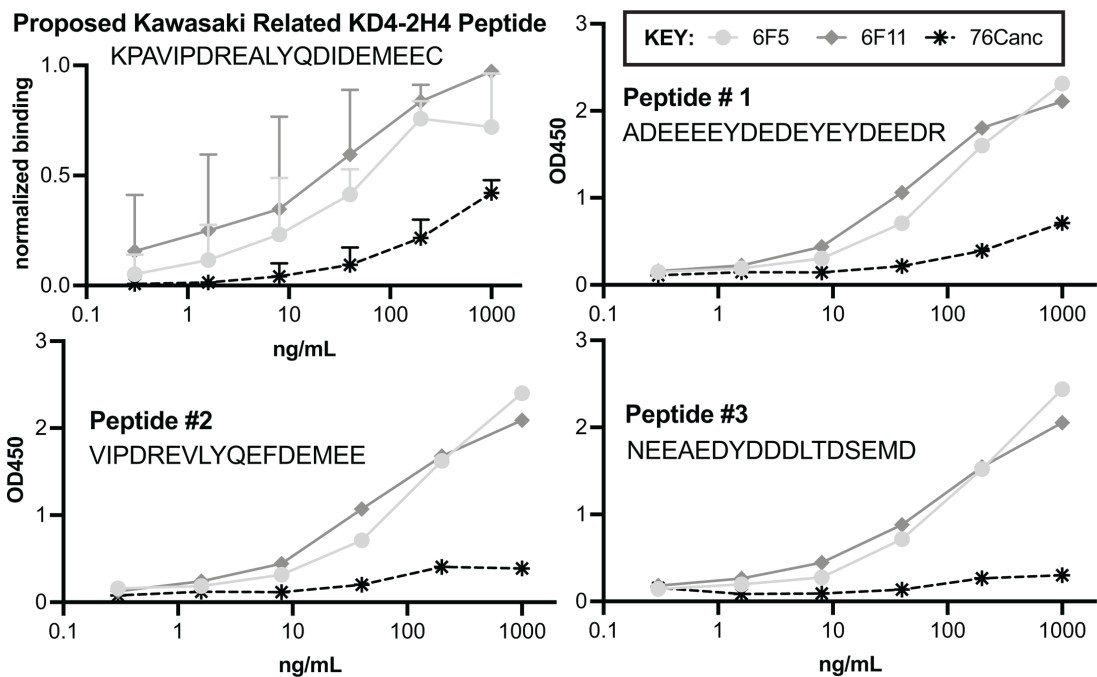

**FIG 4** 76C antibodies all recognize Hepatitis C virus-related peptide implicated in KD. ELISA binding to KD4-2H4 and the top three peptides on our screen were performed using 76Canc (starred), 6F5 (light gray circle), and 6F11 (gray diamond). In the first panel, normalized data of three experiments are shown with means and standard errors, other data being single runs showing consistent patterns for all three peptides.

## DISCUSSION

In this study, we initially sought to discover a protein target that could potentially replicate the epitope (i.e., a mimotope) targeted by the 76C group Abs to be developed for future immunization studies. Surprisingly, a peptide identified by our anti-HIV Abs was highly similar to a peptide implicated in KD. We had initial concern in developing this peptide into a vaccine candidate due to the published findings with KD.

### Relationship to Kawasaki disease

It is unclear how Abs targeting this peptide relate to KD. This peptide is related to a sequence of a non-structural protease, which would be atypical for a significant immune response. There are no direct sequencing studies that show any *Hepacivirus* member is related to KD. New PHIP-seq (27) approaches using overlapping peptides with multiple *Hepacivirus* antigens have also failed to show an association (28, 29). Notably, prior studies have attempted to link CoVs as the cause of KD (30), but as reviewed, there were no significant targeting of CoV-related peptides in our screen. Also, in our prior studies comparing KD to febrile controls, we did not note any specific differences in targeting the Spike proteins of various CoVs, including severe acute respiratory syndrome coronavirus 2 (SARS-CoV-2) (26).

   The Abs that originally identified the KD4-2H4 peptide were derived from plasmablasts. We have shown that KD children have plasmablast responses similar to children responding to an infection (16), so conceptually this is a plausible approach. Antigen specificity has been shown when peripheral plasmablast levels peak, usually 5–10 days after antigen challenge (31, 32). In our prior study, the peak of plasmablasts in KD was on day 5 of fever (16). It's reported that the Abs that originally identified the KD4-2H4 peptide were derived from plasmablasts roughly 2 to 3 weeks into fevers (20). If this KD4-2H4 peptide was identified by such an off-peak plasmablast-derived Ab, it may reflect a target of non-specific background plasmablasts that circulate at low percentages between periods of antigen stimulation. KD4-2H4 did recognize inclusions

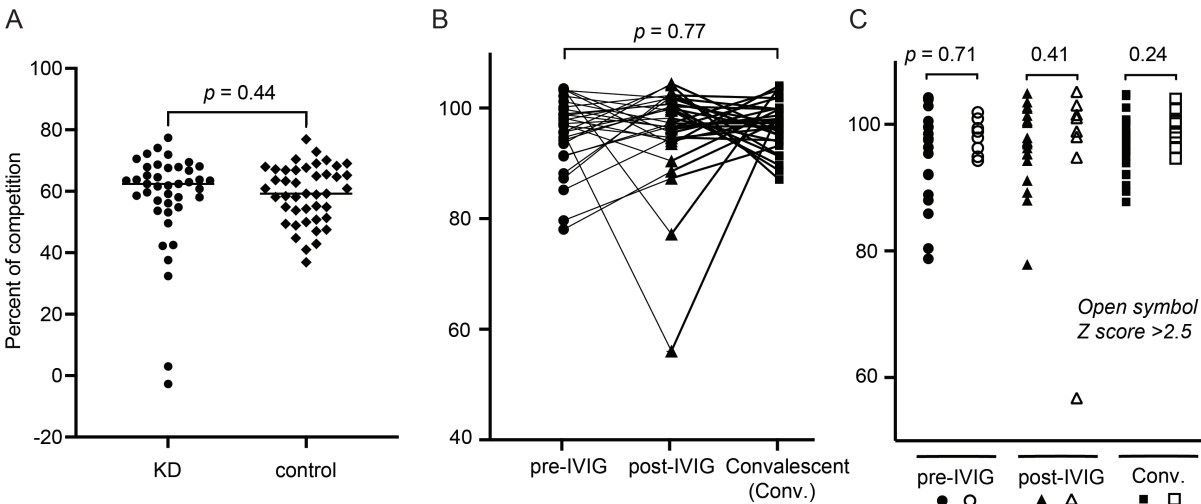

**FIG 5** Humoral immune targeting to Hepatitis C virus-derived peptide does not specifically identify children with KD. (A) Serum at 1:200 was used to compete against biotinylated 6F11 binding to KD4-2H4 peptide (KPAVIPDREALYQDIDEMEEC). This was normalized to background negative competition wells as reading was 0% competition in KD (circle) and controls (diamond). (B) Immune targeting was assessed in serial samples (pre-IVIG—circle, post-IVIG—triangle, convalescent—square) from 30 individuals with KD. (C) Boston scoring for coronary artery aneurysms was used to define Z scores >2.5 (open symbols) as previously published (26). These represent a single experiment due to sample availability.

from bronchial tissue which are posited to indicate viral protein inclusions (20), but could represent inflammation-related aggresomes, or other similar structures as previously reviewed (33). Prior studies using deep sequencing of these inclusions have not revealed significant infection signatures, calling into question their significance (22, 33). Notably, these Abs that targeted KD4-2H4 peptide had variable binding on prior published assays (34), so possibly our competition assay did not fully reflect optimal antigen targeting.

## Mimotope derived from the Hepatitis C virus

Mimotope discovery is purely based on structural homology, so interpretation of specific peptides should proceed cautiously. We were using this study to specifically find a mimotope for an HIV epitope, so the origin of such a peptide may not have any biological relevance. The KD4-2H4 targeting Abs may similarly have no relation to a *Hepacivirus*, but may be targeting a mimotope. In fact, KD4-2H4 binding improved when modifying the *Hepacivirus*-derived peptide in areas that diminished the homology to *Hepacivirus* (34). HCV is associated with autoimmune disorders (35–37) supporting the possibility that this cross-reaction is being driven by an autoimmune reaction. Notably, 10076, the subject from which 76C group Abs was derived, was reportedly negative for HCV (38).

## Other autoimmune targets of germline VH1-02 constructed Ab

Of the human protein targets found on the microarrays, autoAbs to these proteins have not been described in HIV. As many of these contain numerous acidic domains and relatively lower binding specificity in the initial peptide screen, these are likely non-specific reactions. AutoAbs to TMOD1 have been associated with pancreatic cancers (39) and IGA nephropathy (40) but no literature related to KD or HIV was discovered in our review. A number of the HIV bnAbs have been described as having autoimmune potential (33, 41). Prior studies suggest gp41 targeting during initial infection relies predominantly on stimulating memory B cells that have previously been activated by non-HIV-1 antigens. A similar study reverting to germline other gp41 targeting Abs lost HIV reactivity but gained poly-reactivity to various host or gut flora antigens (42). Groups have postulated that germline Abs primed by reactions to commensal bacteria can be stimulated and form the basis for anti-gp41 Ab responses after infection (43).

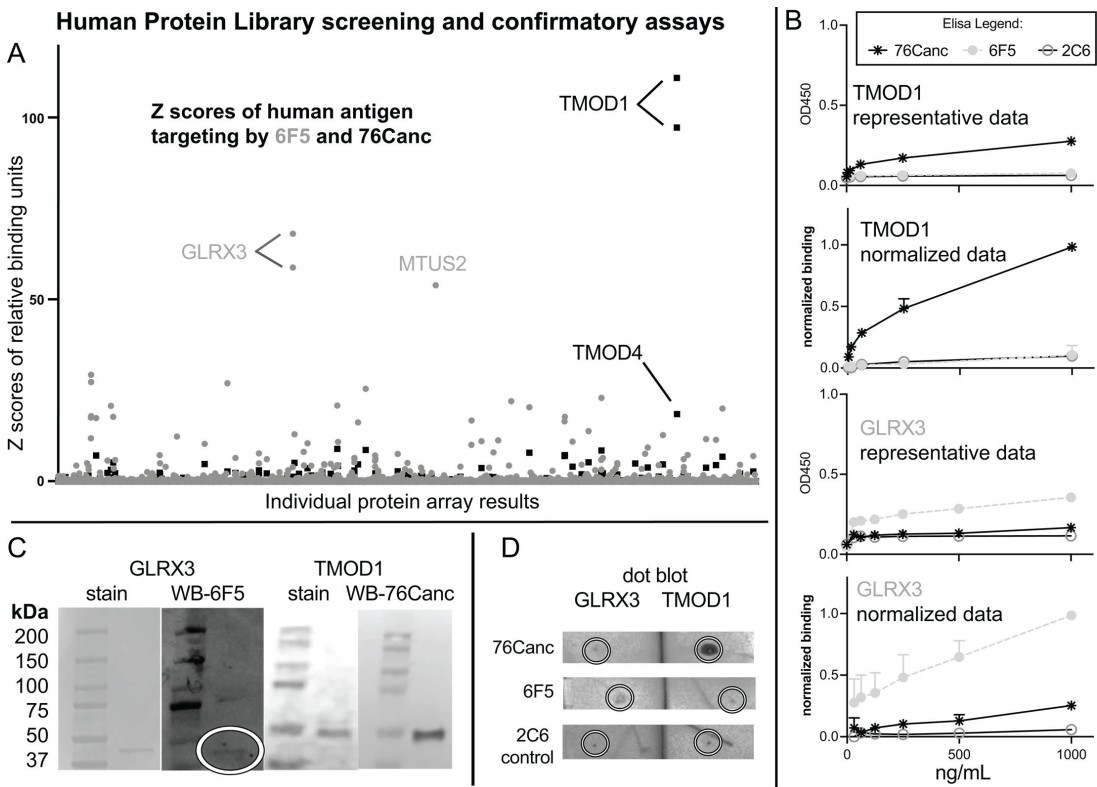

**FIG 6** Binding to yeast displayed human proteins. (A) Results of the full library interrogation for 23059 *Saccharomyces cerevisiae* expressed and purified human proteins (HuProt library, CDI labs) are displayed, for 76Canc (black) and 6F5 (gray). (B) ELISA confirms low-level binding with representative and normalized data of three experiments showing means and standard errors. (C) Western blot (WB) is aligned adjacent to the Coomassie blue stain. (D) Native protein was blotted in a dot blot format with areas of inoculation at the pen marks in the center of the circle. Representative blots are shown of experiments repeated at least twice.

On our screen, the peptides showing highest binding were generally not derived from organisms that would fall into the "gut microbiome" realm (see Table 1; Table S1). It is possible that there is some microbiome dysregulation in both KD and HIV that may explain the cross-reactivity to KD4-2H4.

## Conclusion

Herein we identify a mimotope of a complex epitope that has been associated with functional Abs that associate with long-term non-progression. Since there have been no confirmatory studies supporting an association of HCV with KD, and we herein show no association of serum targeting in our KD samples, we believe this mimotope is a viable candidate to advance to pre-clinical HIV vaccination studies.

## ACKNOWLEDGMENTS

The study was supported by the Wildermuth Research Foundation (M.D.H.) through the Variety Club of Buffalo and NIH R01 AI 125119-01 (M.D.H.); "The role of non-broadly neutralizing antibodies targeting gp41 structural epitopes in long-term non-progression of HIV infection." M.D.H. is a site PI for a Pfizer study, unrelated to the contents of this manuscript. Data available by request.

Conceptualization, M.D.H.; Methodology, M.D.H., S.B., and H.S.; Formal analysis, M.D.H., S.B., and H.S.; Investigation, S.B. and H.S.; Writing original draft, M.D.H.; Review and editing, S.B. and H.S.; Visualization, M.D.H., S.B., and H.S.; Project administration, M.D.H.; Funding acquisition, M.D.H. All authors have read and agreed to the published version of the manuscript.

## AUTHOR AFFILIATION

[1]Department of Pediatrics, University at Buffalo, Buffalo, New York, USA

## AUTHOR ORCIDs

Mark D. Hicar  http://orcid.org/0000-0002-1768-5419

## FUNDING

| Funder | Grant(s) | Author(s) |
| --- | --- | --- |
| National Institute of Allergy and Infectious Diseases | R01 AI 125119-01 | Mark D. Hicar |

## AUTHOR CONTRIBUTIONS

Hakimuddin Sojar, Formal analysis, Investigation, Methodology, Visualization, Writing – review and editing | Sarah Baron, Formal analysis, Investigation, Methodology, Visualization, Writing – review and editing | Mark D. Hicar, Conceptualization, Formal analysis, Funding acquisition, Methodology, Project administration, Visualization, Writing – original draft, Writing – review and editing

## ADDITIONAL FILES

The following material is available online.

### Supplemental Material

**Supplemental File S1 (Spectrum01911-24-s0001.pdf).** TMOD1 alignment with peptides identified from this study Blast and CoBalt were used to show potential acidic rich regions that may account for cross-reactivity.
**Table S1 (Spectrum01911-24-s0002.pdf).** Notable peptides from the screen with 76Canc.

### Open Peer Review

**PEER REVIEW HISTORY (review-history.pdf).** An accounting of the reviewer comments and feedback.

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
