## [Reviewer comments · Microbiology Spectrum]

Microbiology Spectrum

Identification of a mimotope of a complex gp41 Human Immunodeficiency Virus epitope related to a non-structural protein of Hepacivirus previously implicated in Kawasaki disease

Hakimuddin Sojar, Sarah Baron, and Mark Hicar

Corresponding Author(s): Mark Hicar, University at Buffalo Jacobs School of Medicine and Biomedical Sciences

Review Timeline:

Submission Date:	July 30, 2024
Editorial Decision:	September 23, 2024
Revision Received:	December 11, 2024
Accepted:	February 23, 2025

Editor: Jose Martinez-Navio

Reviewer(s): Disclosure of reviewer identity is with reference to reviewer comments included in decision letter(s). The following individuals involved in review of your submission have agreed to reveal their identity: Arden Baylink (Reviewer #2)

Transaction Report:

DOI: <https://doi.org/10.1128/spectrum.01911-24>

Re: Spectrum01911-24 (**Identification of a mimotope of a complex gp41 Human Immunodeficiency Virus epitope related to a non-structural protein of Hepacivirus previously implicated in Kawasaki disease**)

Dear Dr. Mark Daniel Hicar:

Thank you for the privilege of reviewing your work. Below you will find my comments, instructions from the Spectrum editorial office, and the reviewer comments.

Revision Guidelines

Sincerely,
Jose Martinez-Navio
Editor
Microbiology Spectrum

Reviewer #1 (Comments for the Author):

General comments

The manuscript, titled "Identification of a mimotope of a complex gp41 Human Immunodeficiency Virus epitope related to a non-structural protein of Hepacivirus previously implicated in Kawasaki disease," aims to identify structural homologues to the region

of the gp41 protein recognized by the 76C epitope-targeting antibody group, which is present at high levels in the serum of HIV long-term non-progressors. The authors employ an elegant methodology to identify mimotopes by screening an extensive peptide array based on human B-cell epitopes, followed by MEME analysis to elucidate structural properties of consensus targets. Notably, one of the top peptide hits corresponds to a Hepatitis C virus (HCV) derived peptide similar to a peptide implicated in Kawasaki disease. The authors commendably extend the study to explore the potential for autoimmunity in the context of vaccine development.

Methodologically, the manuscript is sound and insightful. However, the findings could have even greater impact if the identified mimotope is validated in immunization studies, demonstrating the capacity to elicit broadly neutralizing antibodies. Moreover, the clarity of the manuscript could be significantly enhanced by providing more relevant background information in the introduction, more explicitly stating the aims from the outset, and ensuring that each results section clearly aligns with those aims. In particular, the abstract could benefit from a clearer articulation of the study's goals and its broader implications.

Specific comments:

Line 48: I suppose the 'highly mutated VH1-02 antibody termed...' means an 'antibody group' rather than an 'antibody'.

Line 87: 'mapping' should read 'mapped'.

The paragraph on VRC01 in the background section is highly unclear.

Lines 200-201: HIV and CoVs are described as type 1 fusion proteins.

Figure 3 does not have A) or B) on the panels.

Line 248: 'Notable other targets' - 'Other notable targets'?

Line 260: 'We did express...' - 'we expressed'

Line 281: What do you mean by 'advanced' specifically in this context?

Lines 301-302: The opening sentence is unclear.

In lines 348-349, "Western blot analysis showed inconsistent resolution of binding (not shown)", these data should be shown in order to assess the results appropriately.

Lines 333-334: How do the authors interpret the lack of binding here compared to the peptide array.

Figure 6 shows binding to human proteins based on yeast display, the authors mention a number of cross-reactions but only the identity of GLRX3 and TMOD1 are disclosed. The rest of the cross-reactions are described as of 'unclear significance', but are not disclosed.

Line 363: 'sequnceing' - sequencing.

Line 387: 'may be related' - related to what?

Reviewer #2 (Comments for the Author):

Sojar and colleagues identify a mimotope that may have therapeutic potential in HIV vaccination studies. Yet, the authors made the surprisingly finding that it is associated with Kawasaki Disease, and undertook a variety of studies to assess whether this could complicate its potential as a viable vaccine candidate. In these studies, the authors do not find any outright risk. This manuscript is generally well-written and provides a novel finding with clinical relevance that will be a meaningful contribution to the field of HIV vaccine research.

My only main concern is that the figures generally do not provide evidence that experiments were performed multiple times, and do not note the n, error bars, or define what the data represent (means, median, etc).

Other comments:

Line 48 and throughout abstract. The authors overuse abbreviations, without defining these abbreviations, and field-specific jargon that make the text inapproachable to a broad audience.

Line 186-189. The authors should clarify that the peptide properties of isoelectric point and hydrophilicity are (I assume) calculated, rather than directly measured. This is important, because there are disagreements in the field about the usefulness and accuracies of these calculations, since there is some dependence on how the experiment is conducted. See: DOI: 10.1002/pro.5560060618

Fig. 3. The legend should specify what assay was used to detect binding, and how this relates to OD450. Otherwise, this plot is not easy to interpret in terms of whether higher values correspond to more binding. Moreover, I believe the y-axis legend should be A450.

Fig. 3, Fig. 4 and others. Many figures do not contain any error bars nor n values. These must be added, as well as a mention as to what the data bar graphs represent (means?).

Line 349. "BLAST alignment did not reveal..." The authors could show this as a panel in the supplemental material.

General comments

The manuscript, titled “Identification of a mimotope of a complex gp41 Human Immunodeficiency Virus epitope related to a non-structural protein of Hepacivirus previously implicated in Kawasaki disease,” aims to identify structural homologues to the region of the gp41 protein recognized by the 76C epitope-targeting antibody group, which is present at high levels in the serum of HIV long-term non-progressors. The authors employ an elegant methodology to identify mimotopes by screening an extensive peptide array based on human B-cell epitopes, followed by MEME analysis to elucidate structural properties of consensus targets. Notably, one of the top peptide hits corresponds to a Hepatitis C virus (HCV) derived peptide similar to a peptide implicated in Kawasaki disease. The authors commendably extend the study to explore the potential for autoimmunity in the context of vaccine development.

Methodologically, the manuscript is sound and insightful. However, the findings could have even greater impact if the identified mimotope is validated in immunization studies, demonstrating the capacity to elicit broadly neutralizing antibodies. Moreover, the clarity of the manuscript could be significantly enhanced by providing more relevant background information in the introduction, more explicitly stating the aims from the outset, and ensuring that each results section clearly aligns with those aims. In particular, the abstract could benefit from a clearer articulation of the study’s goals and its broader implications.

Specific comments:

Line 48: I suppose the ‘highly mutated VH1-02 antibody termed...’ means an ‘antibody group’ rather than an ‘antibody’.

Line 87: ‘mapping’ should read ‘mapped’.

The paragraph on VRC01 in the background section is highly unclear.

Lines 200-201: HIV and CoVs are described as type 1 fusion proteins.

Figure 3 does not have A) or B) on the panels.

Line 248: ‘Notable other targets’ – ‘Other notable targets’?

Line 260: ‘We did express...’ – ‘we expressed’

Line 281: What do you mean by ‘advanced’ specifically in this context?

Lines 301-302: The opening sentence is unclear.

In lines 348-349, “Western blot analysis showed inconsistent resolution of binding (not shown)”, these data should be shown in order to assess the results appropriately.

Lines 333-334: How do the authors interpret the lack of binding here compared to the peptide array.

Figure 6 shows binding to human proteins based on yeast display, the authors mention a number of cross-reactions but only the identity of GLRX3 and TMOD1 are disclosed. The rest of the cross-reactions are described as of 'unclear significance', but are not disclosed.

Line 363: 'sequeing' – sequencing.

Line 387: 'may be related' – related to what?

Sojar and colleagues identify a mimotope that may have therapeutic potential in HIV vaccination studies. Yet, the authors made the surprisingly finding that it is associated with Kawasaki Disease, and undertook a variety of studies to assess whether this could complicate its potential as a viable vaccine candidate. In these studies, the authors do not find any outright risk. This manuscript is generally well-written and provides a novel finding with clinical relevance that will be a meaningful contribution to the field of HIV vaccine research.

My only main concern is that the figures generally do not provide evidence that experiments were performed multiple times, and do not note the n, error bars, or define what the data represent (means, median, etc).

Other comments:

Line 48 and throughout abstract. The authors overuse abbreviations, without defining these abbreviations, and field-specific jargon that make the text inapproachable to a broad audience.

Line 186-189. The authors should clarify that the peptide properties of isoelectric point and hydrophilicity are (I assume) calculated, rather than directly measured. This is important, because there are disagreements in the field about the usefulness and accuracies of these calculations, since there is some dependence on how the experiment is conducted. See: DOI: [10.1002/pro.5560060618](https://doi.org/10.1002/pro.5560060618)

Fig. 3. The legend should specify what assay was used to detect binding, and how this relates to OD450. Otherwise, this plot is not easy to interpret in terms of whether higher values correspond to more binding. Moreover, I believe the y-axis legend should be A450.

Fig. 3, Fig. 4 and others. Many figures do not contain any error bars nor n values. These must be added, as well as a mention as to what the data bar graphs represent (means?).

Line 349. "BLAST alignment did not reveal..." The authors could show this as a panel in the supplemental material.

Identification of a mimotope of a complex gp41 Human Immunodeficiency Virus epitope related to a non-structural protein of Hepacivirus previously implicated in Kawasaki disease

Reviewer Response

We would like to thank the reviewers for their thorough review and for contributing their time to peer review. We have addressed the comments and have made extensive revisions to text and figures as suggested.

Reviewer 1:

“... findings could have even greater impact if the identified mimotope is validated in immunization studies, demonstrating the capacity to elicit broadly neutralizing antibodies.”

Yes, we agree with the reviewer. These studies are currently being pursued and are beyond the scope of the current manuscript. This current study was to assess if preclinical studies should be pursued and this point has been clarified.

“... more relevant background information in the introduction”, more explicitly stating the aims from the outset, and ensuring that each results section clearly aligns with those aims.”

We have edited the background to more thoroughly address the aims of this study. We have also edited each results section to address alignment with the overall aims of the study.

“In particular, the abstract could benefit from a clearer articulation of the study's goals and its broader implications.”

We have revised the abstract to add clarity keeping it under 250 words. Per journal guidelines, we removed headers. We also added an “importance” section as required by the journal that will add clarification as well.

Specific comments:

Line 48: I suppose the 'highly mutated VH1-02 antibody termed...' means an 'antibody group' rather than an 'antibody'.

From prior publications, we have grouped antibodies together due to genetic similarity. This is referring to the specific full title of these antibodies as previously published. We added language to clarify this.

Line 87: 'mapping' should read 'mapped'.

we have made that change, thank you

The paragraph on VRC01 in the background section is highly unclear.

We have revised this section to clarify this background.

Lines 200-201: HIV and CoVs are described as type 1 fusion proteins.

We have edited this for clarification.

Figure 3 does not have A) or B) on the panels.

We adjusted the figure accordingly.

Line 248: 'Notable other targets' - 'Other notable targets'?

edited

Line 260: 'We did express...' - 'we expressed'

corrected

Line 281: What do you mean by 'advanced' specifically in this context?

The targeting of this peptide has been associated with diagnosis by other studies.

Lines 301-302: The opening sentence is unclear.

We revised the sentence to attempt clarification.

In lines 348-349, "Western blot analysis showed inconsistent resolution of binding (not shown)", these data should be shown in order to assess the results appropriately.

We added these blots to figure 6. Binding to GLRX3 is lower affinity than TMOD1, as shown, so resolution of bands was not consistently achieved. Additional recombinant antigen was obtained which showed more consistent results, but still was apparently a low affinity interaction.

Lines 333-334: How do the authors interpret the lack of binding here compared to the peptide array.

We have added language to reflect this comparison. The proteins recognized by recombinant yeast display are not the top *Homo sapien*-derived peptides identified. Peptides may be recognized as mimotopes of structural determinants on proteins or may be due to non-specific binding, likely explaining these differences. We have added language to address this.

Figure 6 shows binding to human proteins based on yeast display, the authors mention a number of cross-reactions but only the identity of GLRX3 and TMOD1 are disclosed. The rest of the cross-reactions are described as of 'unclear significance', but are not disclosed.

We have added language to clarify how significance was defined within this assay and have added the other significant hits to the figure.

Line 363: 'sequencing' - sequencing.

fixed

Line 387: 'may be related' - related to what?

We have expanded this paragraph to clarify this relation.

Reviewer 2:

“This manuscript is generally well-written and provides a novel finding with clinical relevance that will be a meaningful contribution to the field of HIV vaccine research.”

We appreciate the generous comments.

My only main concern is that the figures generally do not provide evidence that experiments were performed multiple times, and do not note the n, error bars, or define what the data represent (means, median, etc).

We have revised some figures (3 and 4) to include normalized data of multiple experiments, instead of representative data. We have clarified errors bars and ranges mean in the legend and have clarified number of runs of data in other experiments.

Other comments:

Line 48 and throughout abstract. The authors overuse abbreviations, without defining these abbreviations, and field-specific jargon that make the text inapproachable to a broad audience.

We have attempted to revise the abstract to make this more accessible to the general audience, and to correspond to the requirements by the journal.

Line 186-189. The authors should clarify that the peptide properties of isoelectric point and hydrophilicity are (I assume) calculated, rather than directly measured. This is important, because there are disagreements in the field about the usefulness and accuracies of these calculations, since there is some dependence on how the experiment is conducted. See: DOI: 10.1002/pro.5560060618

We have clarified this and have highlighted this in our methods.

Fig. 3. The legend should specify what assay was used to detect binding, and how this relates to

OD450. Otherwise, this plot is not easy to interpret in terms of whether higher values correspond to more binding. Moreover, I believe the y-axis legend should be A450.

We have added the biosensor used to the methods section. Per manufacturer, they report absorbance as optical densities.

Fig. 3, Fig. 4 and others. Many figures do not contain any error bars nor n values. These must be added, as well as a mention as to what the data bar graphs represent (means?).

For figure 3 and figure 4, we showed representative results from the experiments. For figure 3, we have revised the figure to use normalized data of the experiments. Figure 4 was originally a single panel, but other peptide results were requested from prior review, but reagents are now exhausted. We put in normalized data from 3 experiments in the first panel. We have clarified errors bars and ranges mean in the legend and have clarified number of runs of data in other locations. We similarly adjusted figure 6.

Line 349. "BLAST alignment did not reveal..." The authors could show this as a panel in the supplemental material.

We added a supplemental figure to illustrate potential acidic rich regions that may be the basis for cross-reactivity.

Re: Spectrum01911-24R1 (**Identification of a mimotope of a complex gp41 Human Immunodeficiency Virus epitope related to a non-structural protein of Hepacivirus previously implicated in Kawasaki disease**)

Dear Dr. Mark Daniel Hicar:

Your manuscript has been accepted, and I am forwarding it to the ASM production staff for publication. Your paper will first be checked to make sure all elements meet the technical requirements. ASM staff will contact you if anything needs to be revised before copyediting and production can begin. Otherwise, you will be notified when your proofs are ready to be viewed.

Sincerely,
Jose Martinez-Navio
Editor
Microbiology Spectrum

Reviewer #1 (Comments for the Author):

Thank you for addressing the concerns and suggestions raised in the initial review. The added clarifications, expanded discussion of background and importance of the study enhance the clarity and rigor of the manuscript.